# Electron-donating amine-interlayer induced n-type doping of polymer:nonfullerene blends for efficient narrowband near-infrared photo-detection

Quan Liu [1,2] ✉, Stefan Zeiske[3], Xueshi Jiang[1,2], Derese Desta[1,2], Sigurd Mertens[1,2], Sam Gielen[1,2], Rachith Shanivarasanthe[1,2], Hans-Gerd Boyen [1,2], Ardalan Armin [3] & Koen Vandewal [1,2] ✉

Inherently narrowband near-infrared organic photodetectors are highly desired for many applications, including biological imaging and surveillance. However, they suffer from a low photon-to-charge conversion efficiencies and utilize spectral narrowing techniques which strongly rely on the used material or on a nano-photonic device architecture. Here, we demonstrate a general and facile approach towards wavelength-selective near-infrared phtotodetection through intentionally n-doping 500–600 nm-thick nonfullerene blends. We show that an electron-donating amine-interlayer can induce n-doping, resulting in a localized electric field near the anode and selective collection of photo-generated carriers in this region. As only weakly absorbed photons reach this region, the devices have a narrowband response at wavelengths close to the absorption onset of the blends with a high spectral rejection ratio. These spectrally selective photodetectors exhibit zero-bias external quantum efficiencies of ~20–30% at wavelengths of 900–1100 nm, with a full-width-at-half-maximum of ≤50 nm, as well as detectivities of >$10^{12}$ Jones.

Solution-processable blends of electron donating (D) and accepting (A) organic semiconductors with high photon-to-charge conversion efficiency have potential for low-cost and high-performance photo-detectors for visible and near-infrared (NIR) wavelengths[1,2]. Especially in the NIR, there is growing interest with applications in, for instance, biometric monitoring[3], night vision imaging, medical diagnostics, and quality control in agriculture[4–7]. In the recent years, synthetic efforts to lower the optical gaps of organic semiconductors has resulted in NIR-absorbing polymers[3,8–13] and nonfullerene acceptors (NFA)[14–18] which, when applied in broadband NIR organic photodetectors (NIR-OPD), reach specific detectivities (D*) on the order of $10^{12}$ Jones, approaching those of commercial silicon or indium gallium arsenide (InGaAs) inorganic detectors at room temperature, albeit in a limited wavelength range, up to 900–1200 nm (see refs. 15, 16). However, for particular applications, spectrally selective NIR detectors instead of broadband OPDs are desired[19]. These include for example wavelength-selective sensing, chemical analysis and emerging artificial intelligence networks[20]. To realize narrowband NIR detection, the predominant strategy that has been widely used in commercial products is to design broadband detectors in combination with an external dichroic prism or optical filters at the photon input side, which inevitably increases the fabrication complexity, detection system size and cost, but also degrades the overall responsivity at the desired detection wavelength. Hence, inherently narrowband high-performance NIR-OPDs are highly desired for the ever-increasing number of applications requiring

[1]Hasselt University, Agoralaan 1, 3590 Diepenbeek, Belgium. [2]IMOMEC Division, IMEC, Wetenschapspark 1, 3590 Diepenbeek, Belgium. [3]Department of Physics, Swansea University, Singleton Campus, Swansea SA2 8PP, UK. ✉e-mail: quan.liu@uhasselt.be; koen.vandewal@uhasselt.be

photon detection and wavelength discrimination in a compact detection system.

In general, four device design strategies have been proposed to achieve a narrowband NIR photoresponse in an organic photodiode: (1) via internal filtering by organic semiconductor thin films with a carefully selected cut-off wavelength[21,22] or external filtering with multi-layered photonic nanostructures[23–25], (2) by using narrowband light-absorbing materials, for example *J*-type aggregated squaraine dyes[26], (3) by intentionally increasing charge-transfer (CT) absorption at a specific wavelength using optical cavity-enhanced architectures[27–31], and (4) by charge collection narrowing (CCN) in a thick junction, exploiting space-charge effect induced by imbalanced transit times of electrons and holes[32]. Among them, the utilization of the narrow absorbing photo-active material is the most straightforward approach, however, the resulting spectral response normally has a large full-width at half-maximum (FWHM) larger than 80 nm. An optical cavity structure employing a thin-metal semi-transparent film or a highly reflective distributed Bragg reflector (DBR) can overcome this and FWHM smaller than 40 nm have been achieved. However, due to weak CT absorption such cavity-OPDs have limited external quantum efficiencies (EQE) at NIR wavelengths, in particular beyond 1000 nm, as well as an undesired angular dependence of the detection wavelength[31,33,34]. The CCN concept demonstrated by Armin, et al. successfully avoids the short-comings of the approach (2) and (3), but it generally requires largely imbalanced charge carrier transport, limiting its applicability as a generic approach for all material systems. This imbalanced transport is often achieved via increasing the drift distance of one type of carriers resulting in low responsivity and response speed[32,35]. In addition, all strategies except (4) mostly rely on precisely controlled device nanostructures, and require D:A bulk-heterojunctions (BHJ) or internal filtering materials with specific optical or electrical properties. It has been a challenge to develop an universal narrowing approach and simultaneously achieve a small FWHM below 50 nm and an EQE higher than 20%, especially for the designed wavelengths in the 900–1600 nm range (see overview of previously reported narrowband NIR-OPDs in Supplementary Table 3).

In this work, we propose a unique approach for wavelength selective charge collection, enabled by n-type doping of the photoactive layer. This results in a screening of the built-in electric field and no extraction of photo-generated charges in the region close to the electron-extracting contact, and a space-charge region (SCR) of 150–250 nm with a high electric field in the vicinity of the top anode. Only weakly absorbed photons reach and are absorbed in this region, resulting in a narrowband response at wavelengths close to the optical gap of the D:A blend. N-doping of typical 500–600 nm thick active layers is simply achieved through applying an electron-donating amine-interfacial layer beneath the polymer: NFA blend films.

This device concept was demonstrated by fabrication of a series of NIR-OPDs with narrowband detecting wavelengths between 900 nm and 1100 nm. We measured peak EQEs around 20% at zero bias and FWHM of ≤50 nm for five D:A combinations, indicating such spectral narrowing approach is universally applicably when high absorption coefficient nonfullerene photoactive materials are used. Moreover, thanks to the relatively low dark current and noise level in nonfullerene OPDs, we measured wavelength-selective detectivities (>1 × 10$^{12}$ Jones) at room temperature, which are higher than that of uncooled InGaAs NIR detectors and close to that of the state-of-the art broadband Si photodetector with detection wavelength, $\lambda_{det}$ < 1050 nm.

## Results
### Design principle of narrowband photo-detection by selectively collecting space charge
Space charge effects, induced by an imbalanced charge transport or unintentional p-type doping[36–38], have previously been reported in the organic solar cell literature to significantly degrade the total collection

of free carriers and overall cell performance, especially in thick-junction devices[38–42]. However, the deliberate doping of an optically thick photodiode opens a new route for achieving narrowband photodetection that has not been explored before. With the aim to obtain near-infrared narrowband photodetectors, in this work we carefully selected a series of efficient NIR-absorbing nonfullerene blends with various optical gaps. The chemical structures of the polymers and low-gap NFAs, as well as the device architectures are shown in Supplementary Fig. 1. Among them, PCE10: COTIC-4F has the most redshifted spectral response with long wavelength absorption edge >1100 nm, which is used as the model active material in the following text.

In general, high-performance OPDs require a blend thickness on the order of several hundreds of nanometers to decrease parasitic shunt currents and thus suppress the dark and noise current level[43]. In addition to that, for instance, a typical 500 nm thick blend film is easily implemented via spin-coating or doctor blading techniques with good homogeneity. Within such a thick-junction device, a majority of the incident photons are absorbed in the part of the active layer near the transparent electrode due to the high absorption coefficients ($\alpha$) of most used D:A blends. This is visualized in Fig. 1a, where we computed the spectrally and spatially resolved photon absorption profile $A(\lambda, x)$ of the PCE10: COTIC-4F blend film in a 500 nm thick inverted device using a transfer matrix method (TMM), which is directly proportional to the simulated distribution of the electric field amplitude $E(\lambda, x)$, but weighted with the optical constants of the photoactive layer[44],

$$A(\lambda, x) = \frac{4\pi}{\lambda} n(\lambda)\kappa(\lambda)|E(\lambda, x)|^2 \tag{1}$$

Here $E(\lambda, x)$ is normalized to the incident electric field, $n(\lambda), \kappa(\lambda)$ are refractive index and extinction coefficient of the active material. Obviously, in the left panel of Fig. 1a, the absorption per unit depth shows two distinctly different regimes in such an optically thick junction: in the range of 550 to 1050 nm corresponding to the high-$\alpha$ regime exhibited in Fig. 1b, the light penetration depth is rather small, and therefore most carriers are photo-generated within the first 200 nm from the front surface. The electric field profile for selected wavelengths are shown in Supplementary Fig. 3, and incident photons at 808 nm and 980 nm typically follow a Beer−Lambert behavior with an exponential decrease with increasing distance from the front surface. In contrast, for the low-$\alpha$ wavelengths ($\lambda$ < 550 nm and $\lambda$ > 1050 nm), the incident light penetrates much deeper in the active layer and charges are generated over the whole volume, as seen in Fig. 1a and Supplementary Fig. 3 (optical filed profiles at 452 nm and 1100 nm). Similar absorption profiles are observed in the high-$\alpha$ and low-$\alpha$ regions for the other studied nonfullerene blends, as shown in Supplementary Fig. 4. When photo-generated carriers are only collected in a zone of about 150 nm from the back contact, a rather high fraction of photons at wavelengths close to the optical gap will produce a photocurrent, while the other wavelengths in the high-$\alpha$ spectral region are rejected (Supplementary Fig. 5a). This results in a narrowband photoresponse with both a high peak value and high spectral rejection ratio (defined as the ratio of the peak responsivity in the detection range to the peak responsivity out of the detection range[21]), prerequisites for high-performing spectrally selective NIR-OPDs.

An extraction zone of about 150 nm close to the back contact can be realized by a redistribution of the electric field into a nearly field-free region and high field region, as illustrated in the energy band diagram in the right panel of Fig. 1a. Such an electric field distribution can be achieved by intentional electrical (n-type) doping of the photoactive layer. Within the field neutral region close to the electron-extracting contact, charge transport of photo-generated carriers is dominated by diffusion of minority holes and charge collection predominantly depends on the diffusion length, $L_D$ of the

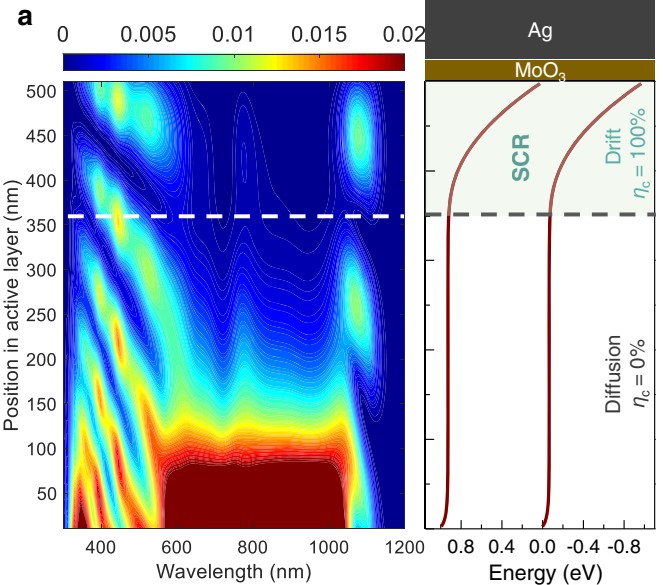

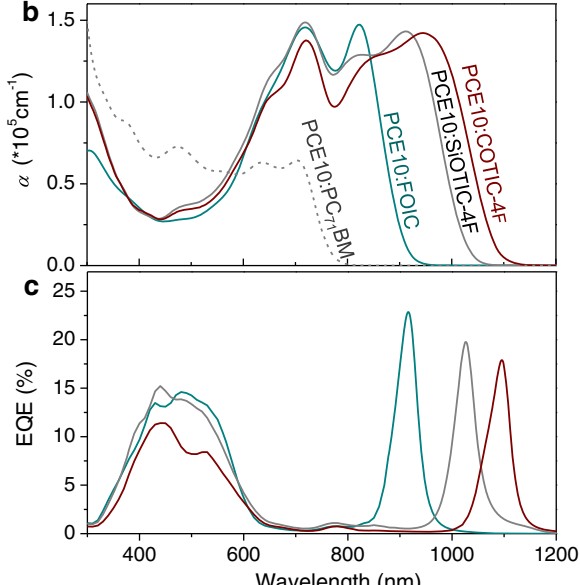

**Fig. 1 | Design principle of near-infrared narrowband photo-detection. a** Left: computed spectrally and spatially resolved energy absorption profile of PCE10: COTIC-4F nonfullerene active layer in an inverted device architecture: ITO (135 nm)/PDIN (15 nm)/PCE10: COTIC-4F (500 nm)/MoO$_3$ (15 nm)/Ag (100 nm). Right: a band diagram at $V = 0$ V in the dark, calculated from a drift-diffusion simulation of the device with a n-doping density of $N_t = 1.88 \times 10^{16}$ cm$^{-3}$ (see simulations details in Supplementary Note 1). The n-type doping causes non-uniform electric field within the active layer of a thick inverted OPD (red). The light green area indicates the space-charge region where the collection of photo-generated charges is rather efficient. **b** Absorption coefficient of the selected near-infrared absorbing PCE10: NFA BHJ blend films with the absorption edges spanning from 900 to 1100 nm. As a reference, PCE10:PC$_{71}$BM fullerene blend is indicated (gray dash line). Note that the absorption coefficient here is calculated from the extinction coefficient $k$ using the relation $\alpha = 4\pi k/\lambda$. **c** Experimental EQE spectra of 500–600 nm thick PCE10: NFA OPDs, measured without an external bias. Note that the complete EQE spectra of PCE10: COTIC-4F OPDs were obtained by scaling the Fourier-transform photocurrent spectroscopy EQE (FTPS-EQE) to the normal EQE, shown in Supplementary Fig. 6.

holes. This makes carriers more likely to recombine in such neutral region, in particular, in a thick-junction (~500 nm) due to the limited $L_D$ for most of efficient nonfullerene photovoltaic D:A blend films ($L_D < 50$ nm)[45]. Most of the internal field is localized in a space-charge region close to the hole extracting top contact, where photo-generated carriers move towards the correct electrode by drift and are effectively collected. Note that such intentional doping induced spectral narrowing approach can be implemented in different device architectures. For a conventional device, p-type doping would be required while for an inverted device configuration with a high work function MoO$_3$/Ag anode as used here, an n-doping strategy is needed. An easy experimental implementation is discussed under the next heading.

Given the large absorption coefficient of NFA blends, almost twice that of typical fullerene-based active layers (see Fig. 1b), an active layer thickness of 500–600 nm is sufficient for achieving a high narrowband EQE with a high spectra rejection ratio when an n-doped device with a fixed SCR width of 150 nm is considered (see thickness dependent absorption profiles in Supplementary Fig. 5c). Whereas for low-$\alpha$ fullerene-based active layers at least >1.0 μm thick is required to obtain such similar spectra rejection ratio (Supplementary Fig. 5e). When the n-doping strategy is applied to the selected NFA blends, we achieved narrowband EQE spectra (Fig. 1c) at NIR wavelengths of 900–1100 nm depending on the absorption onset of the blend, with peak EQE values of about 20% and FWHM ≤ 50 nm (see Lorentz fit to the EQEs in Supplementary Fig. 7). We note that for a typical 500 nm thick junction, a further decrease in the width of the SCR ($w$) or a higher n-doping concentration does not further reduce the FWHM (Supplementary Fig. 5b). However, increasing the active layer thickness does (see FWHM of the simulated narrowband spectra as a function of $w$ or blend thickness in Supplementary Fig. 5c–f). A FWHM of ~30 nm for PCE10:COTIC-4F narrowband OPD is theoretically achievable for an 1100 nm thick blend.

## The role of electron-transporting layer (ETL) in manipulating SCR width and EQE

We selected several electron interfacial materials containing amine groups in their molecular structure, for example, PEIE, PDIN, PFN-Br and their derivatives PDINO and PFN (chemical structures are shown below), and implemented them in PCE10: COTIC-4F (~500 nm) devices. Besides the common belief that amine-interlayers are considered to substantially reduce the work function (WF) of ITO substrates due to the formation of the interface dipoles, some amine groups can donate a lone pair of nonbonding electrons under certain condition to the electron-accepting unit of NFA, resulting in n-type doping of NFA blends.

When using PEIE and PDIN interlayers in devices, we obtained narrowband EQE spectra shown in Fig. 2a, which resemble the calculated spectra depicting the fraction of absorbed photons in a SCR close to the anode (Supplementary Fig. 5a). In contrast, the PFN-Br device exhibited a broadband EQE response from 300 to 1150 nm. As a comparison, a ZnO sol-gel ETL and ETL-free devices were fabricated and similar broadband EQEs as for PFN-Br were observed. Note that the morphology of the active layer is negligible influenced by the electron-transporting interfacial layer below, which is confirmed by atomic force microscope (AFM) and static water contact angle measurements on the different ETL-coated blend films (Supplementary Fig. 8).

We postulate that the origin of this remarkable difference in the shape of the EQE spectra lies in the fact that PEIE and PDIN are able to induce n-type doping in the NFA photoactive layers, while PFN-Br and ZnO are not. We measured current density–voltage ($J$–$V$) characteristics under an 808 nm LED laser diode with an intensity of ~6 mW/cm$^2$. In Fig. 2b, all the devices demonstrated a similar fill factor (FF) around 50%, indicating reasonable charge transport upon excitation at high-$\alpha$ wavelengths. However, a substantial reduction in short-circuit currents were found in PDIN and PEIE devices, in agreement with the EQE spectra. We further measured a well-balanced carrier mobility with a

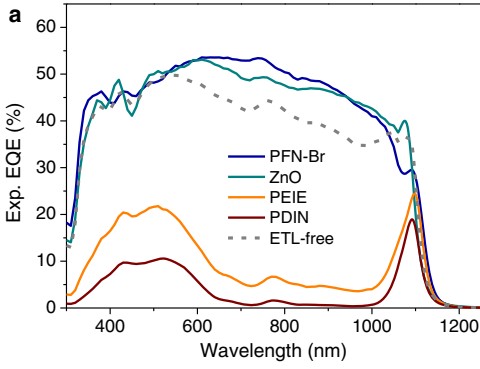

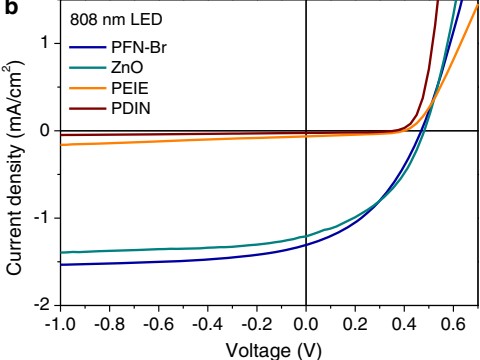

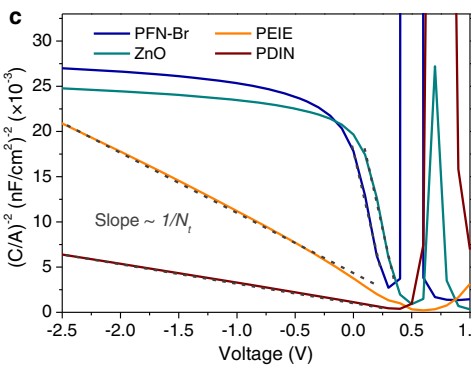

**Fig. 2 | The role of electron-transporting layers (ETL) in manipulating EQE shapes. a** ETL dependent EQE spectra in an inverted 500 nm thick PCE10:COTIC-4F photodiode. For the ETL-free (gray dash line) inverted device, the ITO substrate was not treated by UV Ozone since it would increase the energy barrier for electron extraction. **b** Corresponding $J$–$V$ characteristics measured under a 808 nm LED source at -6 mw/cm$^2$. **c** Mott−Schottky plots of ETL dependent PCE10:COTIC-4F devices, measured at 1 kHz in the dark.

difference of less than half an order of magnitude between electron and hole mobility (see single-carrier mobility measurements in Supplementary Figs. 9 and 10), resulting in a broadband EQE spectrum even if the blend thickness increases to ~1 μm (Supplementary Fig. 11). Also, no obvious slope change in a semi-log plot of light-intensity vs open-circuit voltage ($V_{oc}$) was observed (see comparison of PFN-Br and PDIN devices in Supplementary Fig. 12). This rules out that the suppression of the EQE for the high-$\alpha$ wavelengths when using PEIE and PDIN are related to surface traps[46] or imbalanced mobility induced charge recombination. Moreover, the influence of the surface energy levels of the electron interlayers is also excluded since all the materials on ITO substrates showed similar work function of ~−4.2 to −4.3 eV (see Kelvin probe force microscope measurements in Supplementary Fig. 13).

Instead, we attribute the narrowband EQE response to n-type doping, which we quantify by capacitance−voltage ($C$−$V$) measurements. In Fig. 2c, we observed a quite different Mott−Schottky plots of

$(C/A)^{-2}$ versus applied DC voltage for the two sets of devices. PDIN and PEIE devices show a nearly straight Mott−Schottky plot over a 2.5 V range, which translates in a comparably flat doping profile (Supplementary Fig. 14), indicating that the depth of n-doping is homogeneous throughout the whole NFA blend film. The corresponding slope yields the doping concentration calculated via

$$N_t = -\frac{2}{q\varepsilon_0\varepsilon_r}\left(\frac{\mathrm{d}(C/A)^{-2}}{\mathrm{d}V}\right)^{-1} \qquad (2)$$

where $\varepsilon_0$ is the dielectric constant of the vacuum, $\varepsilon_r$ is the relative dielectric permittivity of the doped active material ($\varepsilon_r$ = 3.5), and $A$ is the device area. We calculated n-doping densities for PDIN and PEIE devices being $1.88 \times 10^{16}$ cm$^{-3}$ and $6.2 \times 10^{15}$ cm$^{-3}$, respectively. The lower doping concentration in PEIE device means a larger width of SCR ($w$), leading to larger collection zone of free carriers and thus a higher EQE achieved than that in the PDIN device (Fig. 2a). This is understood from the equation for the width of the SCR in a doped semiconductor device at an applied bias $V$, given by

$$w = \sqrt{\frac{2\varepsilon_0\varepsilon_r(V_{bi} - V)}{qN_t}} \qquad (3)$$

where $V_{bi}$ is the built-in voltage. For an effective $V_{bi} \approx 1.02$ V roughly determined by the work function difference between cathode and anode contacts, we obtain a width $w = 145$ nm for PDIN device and $w = 252$ nm for PEIE device at short-circuit, respectively. Whereas PFN-Br and ZnO devices show a similar Mott−Schottky plot with an inclination point instead of a clear straight region that is unaffected by the saturation of the capacitance at high voltages. Hence, in this case we cannot determine a reliable value of doping but just an upper limit of $N_A < 7 \times 10^{14}$ cm$^{-3}$ which as compared to the PDIN and PEIE cases can be considered as non-doped devices[47]. This trend is also in accordance with EQE measurements, confirming that the narrowed EQE spectra were obtained due to the formation of positive space charge induced by a n-type doping the low-gap NFA active layers.

Such n-doping between ETL and NFA active layer is also supported by the observation of an improved electrical conductivity (Supplementary Fig. 15). When such electron-donating interfacial layers applied to other NFA blends (PCE10: FOIC, PCE10: SiOTIC-4F, PCE10: IEICO-4F, and PM6:Y6), we obtained very similar EQE results as PCE10:COTIC-4F, as shown in Fig. 1c and Supplementary Figs. 16−18. PEIE or PDIN based devices always show a narrowband EQE and a straight Mott−Schottky plot from the $C$−$V$ measurements (Supplementary Fig. 19). In contrast, devices with ZnO or PFN-Br always exhibit a broadband EQE and no straight region from Mott−Schottky plot is present. Note that PDIN interlayer can be partially washed away by chloroform (CF) during the spin-coating process, hence for the CF-processed NFA blends, PEIE interlayer was used.

In order to further confirm n-type doping and elucidate on its origin in the PEIE or PDIN devices, we performed high-resolution X-ray photoelectron spectroscopy (XPS) on the different types of amine-ETL-coated ITO substrate by tracking the position and shape of the N*1s* peak, shown in Fig. 3a−c. We first consider the N*1s* spectrum (Fig. 3a) of PEIE (processed from H$_2$O) which contains two main peaks from neutral amine nitrogen located at 400 and 400.9 eV, and a relatively small peak at a higher binding energy position of 401.8 eV indicative of protonated amine (N$^+$) in line with previous N*1s* XPS studies[48,49]. The N*1s* peak at the lower binding energy (400 eV) is ascribed to N−CH$_2$−CH$_2$OH tertiary amine due to the attached electron-rich alcohol hydroxyl end groups. To understand which neutral amine group is the main source for electron-donating or n-doping, we processed PEIE from water with different pH values since acidic conditions are considered to promote the protonation of amine while basic conditions

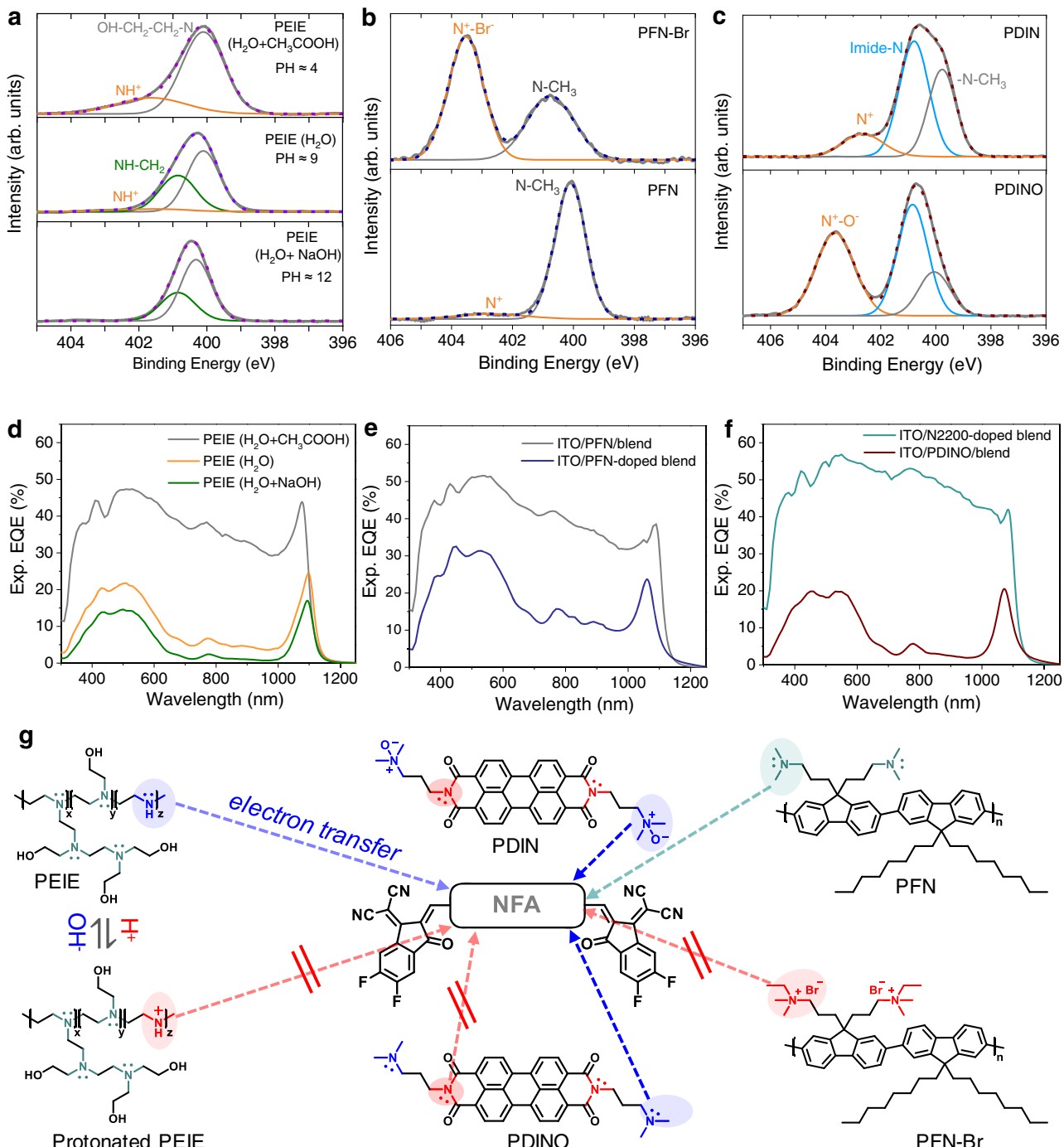

**Fig. 3 | Mechanistic study of n-doping between various ETLs and active layers containing nonfullerene acceptors.** Deconvolution of the N1s peak in XPS spectra of different ETLs coated on ITO substrates: (**a**) PEIE processed using different conditions: 1.0 vol% CH₃COOH and 1.0 vol% NaOH solution (40 mg/mL in H₂O) were added in a 6 mg/mL PEIE (H₂O) solution to tune the pH value to be around 4 and 12, respectively. Note that all the PEIE samples were thermally annealed at 80 °C in a glovebox for 10 min before performing XPS measurements. **b** PFN-Br and PFN. **c** PDIN and PDINO. The experimental N1s curves (bold gray lines) are as-recorded

data after Shirley background subtraction, and the dash lines present their best fits by performing multi-peak fittings. **d**–**f** Measured EQE spectra of an inverted 500 nm thick PCE10:COTIC-4F device using different ETLs as interlayers (as indicated in the figure legend) or directly blended within the active layer. N2200 and PFN are blended directly into the active layer blend with a weight ratio of ~1:50 (wt/wt). **g** Summary of n-doping mechanism of NFA containing blends with the ETLs studied in this work, indicating the possible electron-transfer pathways.

suppress it[49]. Indeed, we observed that the N1s spectrum (Fig. 3a, Top) peaked at 400.9 eV vanishes and shifts to 401.8 eV at low pH conditions of pH ~ 4, implying that the (CH₂)₂-NH- secondary amines in PEIE are fully protonated. As a result, such protonated-PEIE device shows a broadband EQE (Fig. 3d), indicating a significant reduction in n-doping density. On the contrary, protonation of PEIE can be completely

suppressed when processed from a basic solution (PH-12): No N⁺ peak is observed in N1s spectrum, resulting in a narrowband response with lower EQE values indicative of an increased n-doping concentration. These results verify that the number of lone-pair on the secondary amine determines the n-doping ability, and also confirm that protonated amine has no lone-pair to donate free electron to the NFA for

n-doping. The proposed mechanism of electron-transfer pathways between PEIE and NFA is shown in Fig. 3g. In short, to ensure PEIE as an n-dopant, it should be processed from a non-acidic solution.

We then turn to PFN which has tertiary amines on the side-chain. In principle, it can donate an electron from the lone-pair on the nitrogen of the tertiary amines (N$1s$ peak at ~400 eV in Fig. 3b), and previous literature reports have shown its ability to n-dope NFAs[50]. However, the PFN device shows a rather broadband EQE (Fig. 3e) indicating a much weaker doping compared to the (CH$_2$)$_2$-NH- amine group containing PEIE interlayers. Only when we mixed PFN directly in the blend at an 1:50 weight ratio, spectral narrowing is visible. When PFN is converted to PFN-Br, the tertiary amine loses the lone-pair electron and forms a quaternary ammonium salt with Br$^-$, resulting in no free electrons that can be transferred to NFA, and thus a broadband EQE (Fig. 2a). However, this situation changes once a strong electron-accepting unit such as a perylene diimide (PDI) core is present in the molecular structure. This is the case for PDIN and its derivative PDINO, which are also called self-doped n-type materials[51–55]. Self-doping of these materials is verified by the UV–VIS–NIR measurements (Supplementary Fig. 20) showing a broad sub-gap absorption band from 600 to 1600 nm which can be assigned to the polaronic transition (see ref. 53). In these materials, the tethered tertiary amine (PDIN) and ammonium salt (PDINO) can release free lone-pair electrons and electrons from the O$^-$ anions bonded to the quaternary amine, respectively, by intramolecular electron transfer[51,54]. As a result, devices containing such self-doped ETLs show narrowband EQE spectra. The spectra of these devices are even narrower than those with the PEIE interlayer, which indicates a more efficient electron-transfer from ETL to NFA. For a similar PDI acceptor (N2200), which has only imide-N in the molecular structure, the lone-pair electron on the amide (imide-N) is delocalized between the nitrogen and the neighbor oxygen through resonance, and can therefore not contribute to n-type doping. Therefore, the EQE of the device using N2200 blended within the active layer, is broadband (Fig. 3f). Figure 3g summarizes the above discussion: The efficiency of electron transfer or n-doping is highest for self-doping ETLs, followed by secondary amine containing ETLs and tethered tertiary amine containing ETLs. For ammonium salt containing ETLs, the n-type doping effect is the weakest.

## Mechanism of space-charge collection narrowing and EQE modeling

As mentioned above, we here propose an n-doping induced space-charge collection enabling narrowband photo-detection, as illustrated in Fig. 1a. To verify this mechanism, we selected PCE10: COTIC-4F devices employing a PDIN electron-donating ETL and studied the dependence of the BHJ thickness and electric field on the EQE spectra. As shown in Fig. 4a, with decreasing blend thickness, the photo-response (solid lines) in the high-$\alpha$ regime increases and finally, at a thickness of 180 nm, approaches a broadband EQE, in which the peak value near 600 nm is comparable with that of the literature reported optimal EQE using a thin (˂ 100 nm) cell in refs. 14 and 17. It is evident that when the width of SCR approaches the active layer thickness, the electric field spans the full device thickness at short-circuit, resulting in an efficient charge collection for all absorbed photons (high and low $\alpha$). By modeling the EQE spectra assuming the photon absorption profiles determined by TMM and a step-function collection model in which the carrier collection efficiency $\eta_c$ is 100% in the SCR and 0% in the neutral zone[37], we are able to quantitatively explain the variation trend in the shape of EQE spectra with different junction thickness. The device EQE is in that case

$$EQE(\lambda) = A(\lambda,x) \cdot IQE(x) = A(\lambda,x) \cdot \eta \cdot \eta_c(x) \qquad (4)$$

where IQE is the internal quantum efficiency and $\eta$ is a constant factor summarizing the efficiency of all fundamental steps prior to charge collection, that is, exciton diffusion, charge transfer, and charge

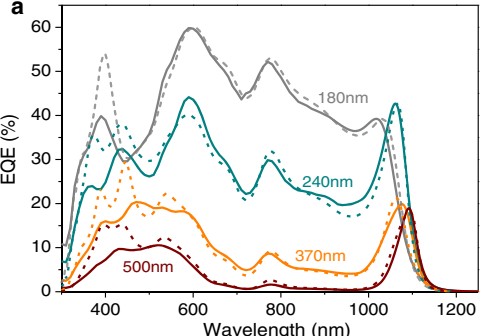

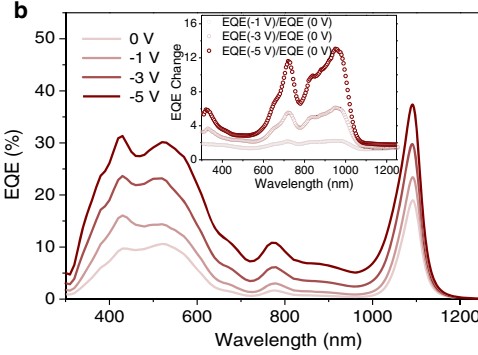

**Fig. 4 | Narrowband EQE modeling and mechanism study. a** Zero bias measured EQE spectra (solid lines) of ITO (135 nm)/PDIN(15 nm)/PCE10:COTIC-4F(180–500 nm)/MoO$_3$ (15 nm)/Ag (100 nm) devices with various active layer thicknesses. The dash lines present the best EQE fits using TMM model. η here is 0.85, determined by scaling the calculated absorption in SCR to match the magnitude of the experimental EQE. **b** EQE spectra of a 500 nm thick PCE10: COTIC-4F inverted device with a PDIN interlayer under different reverse bias. Inset is the EQE change determined by EQE (bias)/EQE (0 V).

separation. Interestingly, with a $\eta$ = 0.85, we found rather good EQE fittings (dash lines) with a width of SCR in the range of 145–155 nm for each case. This implies that the n-doped PCE10:COTIC-4F devices yield a similar width of SCR that is very close to the determined value by Mott–Schottky analysis (Fig. 2c). Once the $w$ is significantly smaller than the blend thickness, surface-generated carriers in the high-$\alpha$ regime easily recombine in the neutral zone, and thus a narrowband photoresponse at the absorption onset of the active material is obtained. Excellent fittings were also found for PCE10: FOIC and PCE10: SiOTIC-4F narrowband devices using PEIE as interlayers (Supplementary Fig. 21). The measured EQEs correspond to a larger $w$ of ~ 250 nm, confirming the weaker doping of PEIE as compared to PDIN (see $N_t$ in Supplementary Table 1). A more precise control of the n-type doping concentration can be realized by adding a small, known amount of such electron-donating interfacial material in the NFA blend. Care must, however, be taken to ensure that the n-dopant does not significantly alter the blend morphology and charge transport properties, otherwise the dark current of the resulting narrowband OPD increases dramatically (Supplementary Fig. 22).

The bias voltage dependence of the EQE spectra of a 500 nm thick PDN interlayer-doped PCE10: COTIC-4F device is shown in Fig. 4b. The EQE increases with applied negative bias due to the enhanced charge extraction and extended width of SCR which can be seen in the energy band diagram plots under bias in Supplementary Fig. 23. The increase in width of SCR also explains why the EQEs in the high-$\alpha$ wavelength range increase more rapidly than those in the low-$\alpha$ wavelength region (see EQE change in the inset of Fig. 4b). Interestingly, even under a large negative bias of −5V, we observed that such narrowband photodiode

does not lose the capability for spectrally selective detection, although a slight larger FWHM and a smaller spectral rejection ratio are seen.

## Device performance of narrowband NIR photodetectors

Having established the mechanism of n-doping enabled spectral narrowing in the NIR photodiodes, we now evaluate their photo-detection performance. Generally, a good OPD should have a high EQE and low

noise current spectral density $i_{noise}$ (with unit of A Hz$^{1/2}$). The latter is determined by multiple factors but at sufficiently high frequencies, $i_{noise} \sim \sqrt{J_d}$ is often valid. Here $J_d$ is the dark current density. A sufficiently thick junction, such as a 500–600 nm thick blend films used in this work, is considered to effectively reduce the influence of surface roughness, pinholes, particles or ITO spikes on the dark current. As shown in Fig. 5a, the measured $J_d$ of the studied narrowband NIR-OPDs

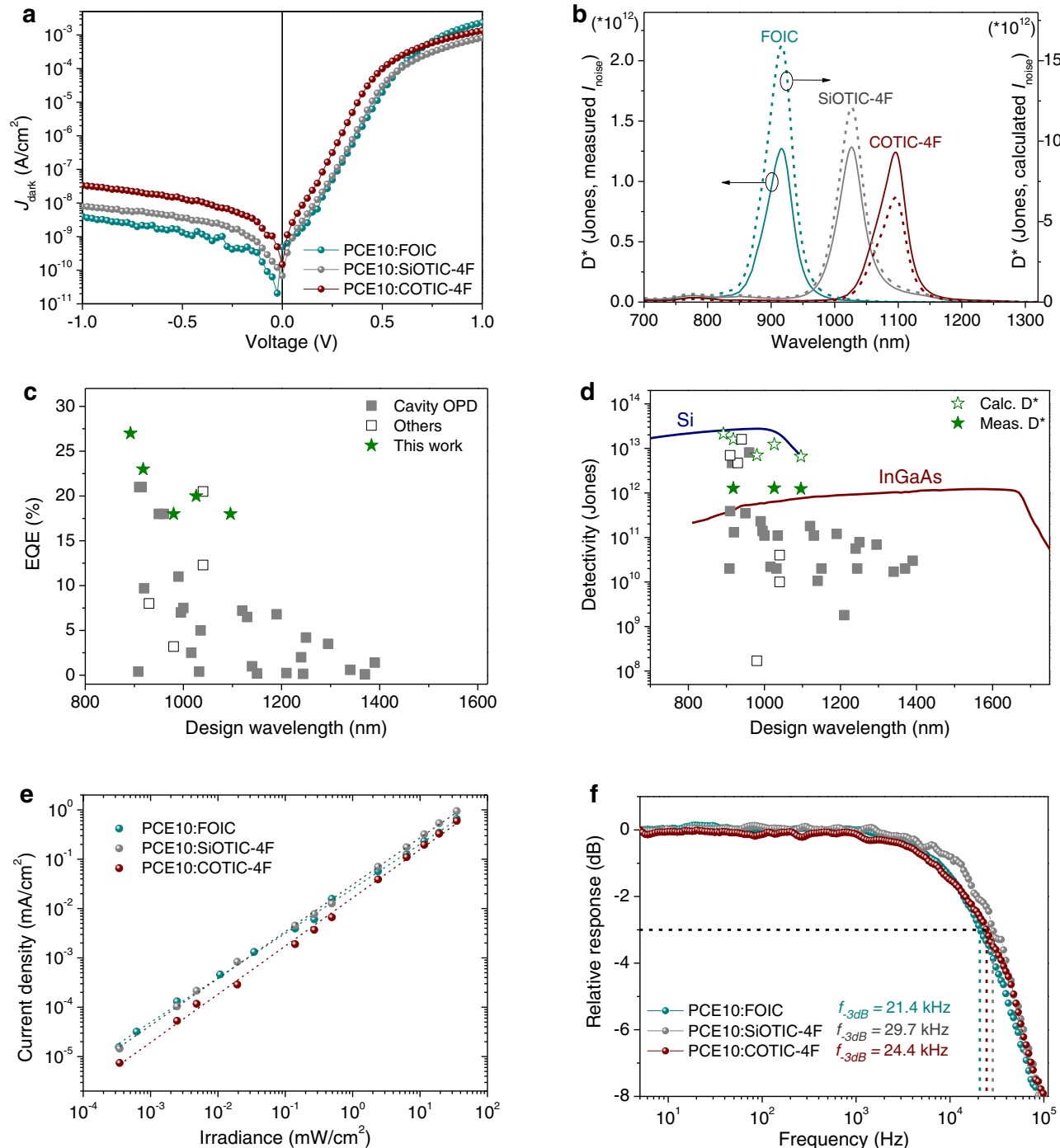

**Fig. 5 | Performance of narrowband near-infrared OPDs. a** Dark current density–voltage ($J$–$V$) characteristics of PCE10: FOIC (600 nm), PCE10: SiTOC-4F (600 nm) and PCE10: COTIC-4F (500 nm) narrowband NIR-OPDs. **b** Corresponding specific detectivity (D*) spectra. Dash lines present the calculated D* at short-circuit condition. (**c**) and (**d**) are the comparison of the achieved performance for the PCE10:NFA narrowband OPDs to other recent reported narrowband NIR-OPDs with the designed $\lambda_{det}$ in the range of 890–1400 nm. The solid lines represent the

commercial inorganic broadband photodetector: Si (blue) and InGaAs (red). **e** Linear dynamic range (LDR) of the optimized narrowband NIR-OPDs measured at 0 V. All the three photodetectors show a linear response of >5 orders of magnitude versus irradiance intensity (*I*). The slope of log(*I*)–log($J_{sc}$) plots for PCE10:FOIC, PCE10:SiOTIC-4F and PCE10:COTIC-4F OPDs are 0.91, 0.95 and 0.97, respectively. **f** Normalized response as a function of input signal frequency. Vertical dash lines mark the cut-off frequencies obtained at −3 dB.

is quite low under reverse bias. For instance, at an operating bias of −1 V the dark currents are on the order of nA/cm². By measuring the $i_{noise}$ spectra shown in Supplementary Fig. 24, a more reliable value for the specific detectivity ($D^*$) of the narrowband NIR-OPDs was calculated.

$$D^* = \frac{e\lambda\sqrt{A} \cdot \text{EQE}}{hc \cdot i_{noise}} \ (\text{Hz}^{1/2}\,\text{cm}\,\text{W}^{-1}, \text{or Jones}) \tag{5}$$

where $\lambda$ is the detection wavelength, $e$ is the elementary charge, device area $A$ here is 0.06 cm², $h$ is Planck's constant, and $c$ is the speed of light in vacuum. As shown in Fig. 5b, the measured peak $D^*$ are above $10^{12}$ Jones (Hz$^{1/2}$ cm W$^{-1}$) at 918 nm, 1026 nm and 1096 nm under zero bias, which is approximately one order of magnitude lower than the calculated $D^*$ at short-circuit condition from $i_{noise} = \sqrt{\frac{4k_B T}{R_{sh}}}$, where $R_{sh}$ is the shunt resistance, derived from the inverse slope of the $J$–$V$ dark curves at 0 V, $k_B$ is Boltzmann constant, and $T$ is the absolute temperature. The calculated performance parameters of narrowband NIR-OPDs were summarized in Supplementary Table 2. When compared to the recently reported narrowband NIR OPDs with designed $\lambda_{det}$ between 890 and 1400 nm, both the EQE values (Fig. 5c) and detectivities (Fig. 5d) we achieved, represent the state-of-the-art for any narrowband detectors (see overview of reported NIR-OPDs in Supplementary Table 3) and the measured $D^*$ is even slightly higher than a typical commercial broadband InGaAs NIR detector. In particular, the PCE10: COTIC-4F OPD achieved a narrowband photo-detection at ~1100 nm beyond the detection limit of a traditional silicon detector. For OPDs a combination of close to 20% EQE, over $10^{12}$ Jones detectivity and FWHM less than 50 nm is rarely reported at such a long wavelength.

Linearity of the photocurrent versus the incident light intensity is another key figure of merit for photodiodes. Linear dynamic ranges (LDRs) at zero bias were also measured and presented in Fig. 5e. All the narrowband photodetectors within 900–1100 nm exhibited at least five orders of magnitude of linear response, expressed as LDR > 100 dB, given by LDR = $20\log(J_{light}/J_{dark})$[56]. Finally, the response speeds of the narrowband OPDs were measured by recording the current response at 0 V. As shown in Fig. 5f, −3dB cut-off frequency ($f_{-3 dB}$) of above 20 kHz were achieved in all the tested OPDs, which are sufficiently fast for a detector array readout. Limited by the light source, the $f_{-3 dB}$ measurements were carried out by using a laser wavelength at 520 nm, however, frequency response should be the same as the light illumination at NIR since carrier transporting in device is independent of illumination wavelength (see Supplementary Fig. 25). Moreover, in agreement with previous reported thick NIR-OPDs[21], we found that the response time of the achieved narrowband NIR-OPDs is dominated by carrier transit time since the calculated resistor–capacitor frequency ($f_{RC} > 620$ kHz for the studied narrowband OPDs) is significantly larger than the measured $f_{-3 dB}$ (see the calculation in Supplementary Table 2).

## Discussion

In conclusion, we propose a generic methodology for spectral narrowing of OPDs in the NIR region based on n-type doping that does not rely on finely controlled nano-photonic structures or selection of internal filtering materials. It is simply achieved in a standard broadband inverted OPD device architecture but with optically thick NFA blends which were intentionally n-doped by electron-donating amine-interlayers. Within a blend film of 500–600 nm, already rather small doping concentrations in the range of $N_t = 10^{16}$ cm$^{-3}$ leads to the creation of a SCR with widths of 150–250 nm, allowing for a high quality narrowband NIR-detection at the absorption onset of the used active materials. A remarkable performance of the fabricated narrowband photodetectors was achieved, exhibiting EQEs of ~20% at NIR region of 900–1100 nm without an external bias, FWHM of ≤50 nm,

high spectral rejection ratios, as well as high detectivities comparable to that of the state-of-the-art broadband organic photodetectors. In addition, from the application point of view, such fabricated photodiodes can be directly used as narrowband near-infrared and broadband visible light dual–model photodetectors[57], due to the absorption nature of the nonfullerene blend films, or easily solution-depositing a 600 nm thick PCE10:PC$_{71}$BM wide-gap material on the front glass substrate to achieve truly visible-blind narrowband NIR-OPDs with negligible responsivity loss (Supplementary Fig. 26). Integrating applications with a diagnostic window extending into the NIR region and without the need for complicated filtering systems, this simple spectral narrowing strategy provides great potential to be applied in the next-generation of high-performance portable and wearable optoelectronics.

## Methods
### Materials and solution preparation

The polymer PTB7-Th and PM6, as well as the nonfullerene acceptors (FOIC, IEICO-4F, Y6, SiOTIC-4F and COTIC-4F) were purchased from 1-material. All photoactive materials were used as received without further purification. The solutions of organic electron-transporting materials: PDIN (1-Material), PEIE (Aldrich, CAS No.: 26658-46-8) and PFN-Br (Ossila) were prepared in methanol (0.5 vol% acetic acid, 4 mg/mL), ultrapure water (6 mg/mL) and methanol (1 mg/mL) at room temperature, respectively. The sol-gel ZnO precursor (0.45 M) was prepared by dissolving zinc acetate dehydrate (Aldrich, 99.9%, 0.5 g) and ethanolamine (Aldrich, 99.5%, 0.14 g) in 2-methoxyethanol (Acros Organics, 99.8%, 5 mL) under vigorously stirring at 60 °C for 2 h in air for hydrolysis reaction[58]. PCE10: COTIC-4F blend solution: PCE10 and COTIC-4F (with a weight ratio of 1:1.5) were dissolved in chlorobenzene (with 2 vol% 1-chloronaphthalene) at 60 °C overnight at a total concentration of 40 mg/mL. PCE10: FOIC (SiOTIC-4F) blend solutions: PCE10 and FOIC (SiOTIC-4F) (with a weight ratio of 1:1.5) were dissolved in chloroform (with 2 vol% 1-chloronaphthalene for SiOTIC-4F blend) at 50 °C for 2 h at a total concentration of 30 mg/mL. All the bulk heterojunction blend solutions were prepared in a N$_2$-filled glovebox with O$_2$ and H$_2$O level ˂ 1 ppm.

### Near-infrared photodetector fabrication

NIR-OPD were fabricated with an inverted device architecture: glass/ITO/ETL/Active layer/MoO$_3$ /Ag. Commercial patterned ITO (Biotain Crystal, 2.5 × 2.5 cm, 15 Ω sq$^{-1}$) substrates were firstly ultrasonic cleaned in sequential in deionized water, acetone, and 2-propanol for 10 min each, followed by 15 min UV-Ozone treatment to form a hydrophilic surface. Afterwards, 10–15 nm of PFN-Br and PDIN were spin-coated on the cleaned ITO in N$_2$-filled glovebox at 3000 rpm without thermal annealing, while 30 nm of ZnO and 10 nm of PEIE (H$_2$O) were spin-coated at 5000 rpm in air and ZnO films need additional thermal annealing at 200 °C for 20 min in atmosphere. Then, ~500 nm thick PCE10: COTIC-4F films were spin-cast on the as-prepared electron-transporting layers in the glovebox from a warm solution (80 °C) at 970 rpm, followed by 5 min heating at 80 °C on a hotplate to remove the residual solvent. ~ 550–600 nm of PCE10: FOIC (SiOTIC-4F) films were formed from their room temperature solutions at 1100 rpm, SiOTIC-4F samples being 110 °C annealed for 10 min. The photoactive layer thickness was monitored by a Bruker Veeco Dektak XT profilometer. Finally, the MoO$_3$ (15 nm) hole-transporting layer and the Ag (100 nm) electrode were sequentially deposited on the active layer through a shadow mask by thermal evaporation (<5 × 10$^{-6}$ mbar) with an area of 0.06 cm². All the electro-optical measurements on the devices were performed in an inert box, except the niose and frequency response measurements carried out at Swansea University, the freshly fabricated devices were encapsulated with glass slides using an UV-curable epoxy (DELO-LP655) in the glovebox and measured in air.

## Dark/ photocurrent and LDR measurement

$J–V$ curves (forward scan with a step of 25 mV) were recorded using a Keithley 2400 Source Meter under AM1.5 1-sun illumination provided by a solar simulator (Newport 91195 A) with an intensity equivalent to 100 mW cm$^{-2}$, which was calibrated with a Silicon reference cell. Dark currents were measured using a Keithley 2400 by averaging the current over time for each voltage step. The devices were fully covered with aluminum foil and measured under room dark condition. An 840 nm long-pass filter filtered solar simulator used as the illumination source with a series of neutral density filters purchased from Thorlabs for the LDR measurements.

## EQE and optical simulation

The EQE spectrum for each OPD was measured using a home-made setup under chopped (135 Hz) monochromatic illumination from a Xe lamp (100 W, Newport) modulated by Cornerstone™ 130 Monochromator and an optical wheel chopper. The generated photocurrent from the solar cells was amplified with a Stanford Research System Model SR830 lock-in amplifier, and a calibrated Si photodiode with known spectral response was used as a reference. The spatially resolved absorbed energy profile and optical field distributions in the photodiodes were simulated using a computer code based on the transfer matrix method. The optical constants (refractive index and extinction coefficient) of all the photoactive materials and ETL involved (see Supplementary Fig. 2) were determined based upon measured transmission spectra in combination with an iterative, reverse transfer matrix model as described previously[59]. Optical constants of the other layers were taken from previously published work[60].

## XPS and capacitance characterization

XPS measurements were performed on a commercial laboratory-based HAXPES-lab system (manufacturer: Scienta Omicron) equipped with 2 independent photon sources (Ga-kα, Al-kα). For the experiments presented here, monochromatized Al-kα photons were used for photoexcitation (XM1000, photon energy 1486.7 eV, power 300 W). The binding energy scale was calibrated by means of a Au reference sample, setting the Au-4$f_{7/2}$ core level position to 84.0 eV. The $C–V$ measurements were performed in the frequency range of 100 Hz to 1 M Hz, from −3 V to +1 V, using an Agilent E4980A precision LCR meter. All the measurements were performed in the dark at room temperature.

## Noise and frequency response measurement

Noise spectral density (NSD) spectra were obtained by recording the dark currents as a function of time and applying a fast Fourier transform (FFT) algorithm. A semiconductor device analyzer (Keysight B1500A) was used for dark current measurements at different bias voltages. The photodetectors were mounted in an electrical-shielded sample holder (Faraday cage) to minimize electrical noise. The frequency response speeds of the narrowband NIR photodetectors were recorded by using a Network Analyzer (Keysight E5071C ENA Vector Network Analyser) in transmission mode modulating an Oxxius L6Cs laser ($\lambda_{exc}$ = 520 nm) with frequencies ranging from 1 Hz to 1 MHz.

## Data availability

All the data supporting the findings of this study are included within the published Article and its Supplementary Information files. Source data are provided with this paper.

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

## Acknowledgements

The authors thank Guy. Brammertz at IMEC for capacitance measurements. We also thank the Research Foundation Flanders (FWO Vlaanderen) for continuing financial support (projects G0D0118N, G0B2718N, 1S50820N, 11D2618N), as well as the European Research Council (ERC, grant agreement 864625). Q.L. acknowledges financial support from the European Union's Horizon 2020 research and innovation program under the Marie-Curie grant agreement no. 882794. S.G. acknowledges the FWO for his Ph.D fellowship. H.-G.B. and D.D. are very grateful to FWO for funding the HAXPES-lab instrument within the HERCULES

program for large research infrastructure of the Flemish government. A.A. acknowledges support from Sêr Cymru II Program through the European Regional Development Fund and the Welsh European Funding Office.

## Author contributions

Q.L. conceived the project, designed and performed the core of the experimental work; S.Z. performed noise and response speed measurements under the supervision of A.A.; X.J. calculated energy band diagrams with a drift-diffusion model; D.D. and H.G.B. carried out the XPS measurements and processed the related data; S.M. helped with *J–V* and LDR measurements; S.G. helped with noise measurement and noise data analysis; R.S. performed KPFM, contact angle and AFM measurements; Q.L. and K.V. wrote the manuscript with the assistance from all other authors.

## Competing interests

The authors declare no competing interests.
