## [Peer Review File · Nature Communications]

Electron-donating amine-interlayer induced n-type doping of polymer:nonfullerene blends for efficient narrowband near-infrared photo-detectionREVIEWER COMMENTS

Reviewer #1 (Remarks to the Author):

This manuscript reports that protonated amine interlayer induces an electrical doping with polymer:nonfullerene active layer and therefore yields narrowband NIR photodetectors with 500-600 nm active layers. The PEIE/PDIN leads to narrowband spectra while ZnO/PFN-Br doesn't. The phenomenon is ascribed to the positive space charge induced by the doping. The finding is interesting and the explanation is reasonable. I would suggest its publication in Nat Commun. The following comments should be elaborated:

1. Why the amines have to be protonated to get the effective doping and narrow the spectral response? Please clarify it in the manuscript.
2. The authors stated the carrier density of their n-doping is 10^{16} or 17 . How does the carrier density correlate to the spectral band? Please add comments on it.
3. There are self-doping ETL in the literature. Will they help to obtain the narrow band?
4. As for the doping, what depth is the doping between the ETL and the active layer? More evidences are needed to confirm the electrical doping, i.e., increase of electrical conductivity?
5. If the top electrode is transparent (thin silver for example), what will the spectral response be when illuminated from the HTL side?
6. For ZnO or PFN-Br ETL, will the spectral response become narrow if the thickness of the active layer further increases to 1 micrometer?
7. Is it possible to remove the PDIN or PEIE (without ETL) to obtain such narrow response?

Reviewer #2 (Remarks to the Author):

Dear Authors and Editor:

The manuscript by Liu et al. entitled "Protonated amine-interlayer induced n-type doping of polymer:nonfullerene blends for efficient narrowband near-infrared photo-detection" has presented a new detector design to achieve narrowband near-infrared response. The design involves incorporating amine interlayers near the anode. This approach is novel and less materials dependent and has been shown to work for 3 different blends, demonstrating the generalizability of this concept. While the device performance is excellent, there are some clarification requests as listed below:

1. The paper has shown that doping concentration is significantly different with the use of PEIE and PDIN interface. Would there be a knob to change the doping concentration more precisely in the future, so the FWHM of the photoresponse can be further reduced? Or, from the simulations, can we extrapolate how N_t or depletion width in the supplemental table S1 will change the FWHM in EQE?
2. In the inset of Fig 4a, there is a dip in the noise spectra in the beginning. Would the authors explain why it is the case? It is not typical and opposite to the usual $1/f$ trend.
3. For the color map in Fig 1a, is the color map on the left in arbitrary units? The scale there is somewhat different from the Supplement calculations in Fig S5 where the calculation absorption in a.u. are ~ 0.2 , but the color maps in Fig 1a and supplemental Fig. S4 is 10x smaller in 0.02.
4. There are a lot of acronyms, making it somewhat difficult to follow, and there is one acronym SCCN that was undefined on p10, but I think it's spectral charge collection narrowing? Anyways, it might be more readable if some acronyms are spelled out in words if possible.

Reviewer #3 (Remarks to the Author):

The manuscript of Liu and coworkers reports narrowband NIR organic photodetectors with a full-width-at-half-maximum of around 50 nm and specific detectivities of $>10^{12}$ Jones. These results are achieved by the n-doping of the active layer induced by protonated amine-based electron transporting layers. The authors pushed further the charge collecting narrowing method developed by Armin et al., and found that a small doping concentration creates a space charge region with widths of 150-250 nm, which leads to narrowband NIR-detection at the absorption onset of the active layer. I have some concerns about the n-doping effect, which are listed below. However, I believe these findings are important for the community and can help the development of NIR OPD. For this reason, I support the publication of this manuscript in nature communications upon replies to the following questions.

- 1) The authors claimed that the n-doping effect and consequentially the SCR is induced by the ETL. The authors supported this statement by indicating that trap-assisted recombination is not playing a role by carrying out light-intensity JVs (Figure S8). I believe that further measurements are necessary to exclude trap-assisted recombination, i.e. charge extraction and transient photovoltage measurements.
- 2) In addition, a common method to study doping is EPR. I strongly recommend the authors to carry out this measurement.
- 3) The morphology of the active layer can be drastically influenced by the layer below. I suggest performing contact angle measurements for active layers coated on different ETL to exclude this variable.
- 4) In some figures, the active layer thickness is missing.

Reviewer #1 (Remarks to the Author):

This manuscript reports that protonated amine interlayer induces an electrical doping with polymer:nonfullerene active layer and therefore yields narrowband NIR photodetectors with 500-600 nm active layers. The PEIE/PDIN leads to narrowband spectra while ZnO/PFN-Br doesn't. The phenomenon is ascribed to the positive space charge induced by the doping. The finding is interesting and the explanation is reasonable. I would suggest its publication in Nat Commun. The following comments should be elaborated:

1. Why the amines have to be protonated to get the effective doping and narrow the spectral response? Please clarify it in the manuscript.

This comment is very relevant and gave us the chance to re-consider the role of protonated-amine in narrowing the spectral response. To better understand the doping mechanism, we did more supplementary experiments (XPS and EQE) and reconsidered our understanding of the function of the amine protonation. In fact, once the amine is fully protonated, no lone-pair electron on the nitrogen can participate in electron transfer to the NFA and thus no n-doping effect would occur. Only amine-ETL with non-tethered lone-pair electrons or electrons from quaternary ammonium anions can induce an n-type doping of non-fullerene active layers. We moved Fig. 2d and Fig.S16 to the revised Fig.3 and re-wrote this section. Please see the revised Fig.3 and relevant discussion in the revised manuscript (pages 9-11),

“In order to further confirm n-type doping and elucidate on its origin in the PEIE or PDIN devices, we performed high-resolution X-ray photoelectron spectroscopy (XPS) on the different types of amine-ETL coated ITO substrates and track the position and shape of the N1s peak, shown in Fig. 3a-c. We first consider the N1s spectrum (Fig.3a) of PEIE (processed from H₂O) which contains two main peaks from neutral amine nitrogen located at 400 and 400.9 eV, and a relatively small peak at a higher binding energy position of 401.8 eV indicative of protonated amine (N⁺) in line with previous N1s XPS studies^{48,49}. The N1s peak at the lower binding energy (400 eV) is ascribed to N-CH₂-CH₂OH tertiary amine due to the attached electron-rich alcohol hydroxyl end groups. To understand which neutral amine group is the main source for electron-donating or n-doping, we processed PEIE from water with different pH values since acidic conditions are considered to promote the protonation of amine while basic conditions suppress it⁴⁹. Indeed, we observe that the N1s spectrum (Fig. 3a, Top) peaked at 400.9 eV vanishes and shifts to 401.8 eV at low pH conditions of pH ~ 4, implying that the (CH₂)₂-NH- secondary

amines in PEIE are fully protonated. As a result, such protonated-PEIE device shows a broadband EQE, indicating a significant reduction in n-doping density. On the contrary, protonation of PEIE can be completely suppressed when processed from a basic solution (PH~12): No N⁺ peak is observed in N1s spectrum, resulting in a narrowband response with lower EQE values indicative of an increased n-doping concentration. These results verify that the number of lone-pair on the secondary amine determines the n-doping ability, and also confirm that protonated-amine has no lone-pair to donate free electron to the NFA for n-doping. The proposed mechanism of electron transfer pathways between PEIE and NFA is shown in Fig. 3g. In short, to ensure PEIE as an n-dopant, it should be processed from a non-acidic solution.

We then turn to PFN which has tertiary amines on the side-chain. In principle, it can donate an electron from the lone-pair on the nitrogen of the tertiary amines (N1s peak at ~400 eV in Fig. 3b), and previous literature reports have shown its ability to n-dope NFAs⁵⁰. However, the PFN device shows a rather broadband EQE (Fig. 3e) indicating a much weaker doping compared to the (CH₂)₂-NH- amine group containing PEIE interlayers. Only when we mixed PFN directly into the blend at an 1:50 weight ratio, spectral narrowing is visible. When PFN is converted to PFN-Br, the tertiary amine loses the lone-pair electron and forms a quaternary ammonium salt with Br⁻, resulting in no free electrons that can be transferred to NFA, and thus a broadband EQE (Fig. 2a). However, this situation changes once a strong electron-accepting unit such as a perylene diimide (PDI) core is present in the molecular structure. This is the case for PDIN and its derivative PDINO, which are also called self-doped n-type materials^{51,52,53,54,55}. Self-doping of these materials is verified by the UV-VIS-NIR measurements (Fig.S20) showing a broad sub-gap absorption band from 600 to 1600 nm which can be assigned to the polaronic transition (see ref. 53). In these materials, the tethered tertiary amine (PDIN) and ammonium salt (PDINO) can release free lone-pair electrons and electrons from the O⁻ anions bonded to the quaternary amine, respectively, by intramolecular electron transfer^{51,54}. As a result, devices containing such self-doped ETLs show narrowband EQE spectra. The spectra of these devices are even narrower than those with the PEIE interlayer, which indicates a more efficient electron-transfer from ETL to NFA. For a similar PDI acceptor (N2200), which has only imide-N in the molecular structure, the lone-pair

electron on the amide (imide-N) is delocalized between the nitrogen and the neighbour oxygen through resonance, and can therefore not contribute to n-type doping. Therefore, the EQE of the device using N2200 blended within the active layer, is broadband (Fig. 3f). Fig. 3g summarizes the above discussion: The efficiency of electron transfer or n-doping is highest for self-doping ETLs, followed by secondary amine containing ETLs and tethered tertiary amine containing ETLs. For ammonium salt containing ETLs, the n-type doping effect is the weakest.

Cited references:

⁵⁰ Kang, Q. et al. Significant influence of doping effect on photovoltaic performance of efficient fullerene-free polymer solar cells. *J. Energy Chem.* 43, 40–46 (2020)

⁵¹ Russ, B. et al. Tethered tertiary amines as solid-state n-type dopants for solution-processable organic semiconductors. *Chem. Sci.* 7, 1914–1919 (2016).

⁵² Wu, Z. et al. n-Type Water/Alcohol-Soluble Naphthalene Diimide-Based Conjugated Polymers for High-Performance Polymer Solar Cells. *J. Am. Chem. Soc.* 138, 2004–2013 (2016).

⁵³ Kang, Q. et al. A printable organic cathode interlayer enables over 13% efficiency for 1-cm² organic solar cells. *Joule*, 3, 227-239 (2019).

⁵⁴ Zhou, D. et al. N-Type Self-Doped Hyperbranched Conjugated Polyelectrolyte as Electron Transport Layer for Efficient Nonfullerene Organic Solar Cells. *ACS Appl. Mater. Interfaces*, 13, 50187-50196 (2021).

⁵⁵ Wang, Z. et al. Self-Doped, n-Type Perylene Diimide Derivatives as Electron Transporting Layers for High-Efficiency Polymer Solar Cells. *Adv. Energy Mater.* 7, 1700232 (2017).

In addition, the title and the relevant sentences in abstract, introduction, and conclusion have been revised accordingly.

Title: “~~Electron-donating~~ amine-interlayer induced n-type doping of polymer:nonfullerene blends for efficient narrowband near-infrared photo-detection ”

Abstract: “nonfullerene blends. We show that a ~~protonated~~-~~electron-donating~~ ~~electron~~ amine-interlayers can induce ”

Introduction: “layers is simply achieved through applying an ~~protonated~~ ~~electron-donating~~ amine-interfacial layer ”

Conclusion: “were intentionally n-doped by ~~electron-donating~~ amine-interlayers. ”

Fig. 3 Mechanistic study of n-doping between various ETLs and active layers containing non-fullerene acceptors. Deconvolution of the N1s peak in XPS spectra of different ETLs coated on ITO substrates: (a) PEIE processed using different conditions: 1.0 vol% CH₃COOH and 1.0 vol% NaOH solution (40mg/mL in H₂O) were added in a 6mg/mL PEIE (H₂O) solution to tune the pH value to be around 4 and 12, respectively. Note that all the PEIE samples were thermally annealed at 80 °C in a glovebox for 10 min before performing XPS measurements. (b) PFN-Br and PFN. (c) PDIN and PDINO. The experimental N1s curves (bold grey lines) are as-recorded data after Shirley background subtraction, and the dotted lines present their best fits by performing multi-peak fittings. (d)-(f) Measured EQE spectra of an inverted 500 nm-thick PCE10:COTIC-4F device using different ETLs as interlayers (as indicated in the figure legend) or directly blended within the active layer. N2200 and PFN are blended directly into the active layer blend with a weight ratio of ~1:50 (wt/wt). (g) Summary of n-doping

mechanism of NFA containing blends with the ETLs studied in this work, indicating the possible electron-transfer pathways.

2. The authors stated the carrier density of their n-doping is 10^{16} or 10^{17} . How does the carrier density correlate to the spectral band? Please add comments on it.

Doping concentration is related to the width of space charge region (SCR) according to the equation 3 in the main text, $w = \sqrt{\frac{2\epsilon_0\epsilon_r(V_{bi}-V)}{qN_t}}$. Please keep in mind that only the carriers in the SCR can be collected in the doped device and contributed to EQE. The second reviewer asked a similar question (see comment 1). We revised Fig. S5, and simulate how the doping concentration (or the width of SCR) and active layer thickness affect the FWHM of the narrowband response.

3. There are self-doping ETL in the literature. Will they help to the obtain the narrow band?

From the above discussion in the comment 1, we believe that literature reported self-doped ETLs (most of them are derivatives of PDIN, see ref. 51-55) can induce n-type doping of NFA blends. So, with a sufficient thick junction (~ 500 nm), self-doped ETLs will indeed help to obtain a narrowband spectral response.

4. As for the doping, what depth is the doping between the ETL and the active layer?

The depth of doping is homogeneous throughout the whole blend film, as indicated by the Mott-Schottky analysis. Please see the apparent doping distribution derived via Mott-Schottky analysis in Figure S14 in the revised Supplementary Information.

Figure. S14 Apparent n-doping density N_{Ms} as a function of active layer position x , for PCE10: COTIC-4F inverted devices on different ETL substrates. The main junction is at the MoO₃/Ag back contact, Here $x = d - \epsilon A/C$, and d is the active layer thickness (500 nm).

Also, a relevant comment is added in the revised manuscript (page. 8):

“Mott–Schottky plot over a 2.5 V range, which translates in a comparably flat doping profile (Fig. S14), indicating that the depth of n-doping is homogeneous throughout the whole blend film. The corresponding slope yields the doping concentration calculated via”

More evidences are needed to confirm the electrical doping, i.e., increase of electrical conductivity?

As suggested by the reviewer, we performed electrical conductivity measurement on the ETL-dependent devices in a configuration of ITO/ETL/PCE10:COTIC-4F (500 nm)/Al, which is an electron-only device. The measured J-V curves are shown below. Indeed, doped devices (PDIN, PEIE) have a higher current density than non-doped devices (PFN-Br, ZnO), indicating improved electrical conductivity by n-doping.

In the revised manuscript (page 9):

“Such n-doping between ETL and NFA active layer is also supported by the observation of an improved electrical conductivity (Fig. S15).”

Figure S15. Semi-logarithmic current density versus voltage characteristics of ETL-dependent electron-only devices, measured in the dark.

5. If the top electrode is transparent (thin silver for example), what will the spectral response is when illuminated from the HTL side?

We expect that the spectral response would be broadband when illuminated from the HTL side since the majority of carriers are in that case generated near the semi-transparent thin-Ag anode where a large electric field is present. To prove it, we fabricated such thin-Ag device with the structure of ITO/PDIN/PCE10:COTIC-4F (500 nm)/MoO₃/Ag (15 nm). When illuminating the PDIN-device from the anode side, the measured EQE turns to be broadband and the shape follows the transmission of thin-Ag electrode, as shown in the below Figure.

6. For ZnO or PFN-Br ETL, will the spectral response become narrow if the thickness of the active layer further increases to 1 micrometer?

To answer the reviewer’s question, we fabricated such thick junction ZnO-device close to 1 micrometer (measured thickness is ~950 nm) with the PCE10: COTIC-4F active material, but the spectral response stays broadband indicative of well-balanced e/h mobilities. please see the measured EQE below. (Fig. S11 in the revised Supplementary Information).

In the revised manuscript (page 7):

“measurements in Figs. S9 and S10), resulting in a broadband EQE spectrum even if the blend thickness increases to ~1 μm (Fig. S11).”

Figure S11. Measured EQE of ITO/ZnO/PCE10: COTIC-4F/MoO₃/Ag device with an ~ 1 μm -thick active layer.

7. Is it possible to remove the PDIN or PEIE (without ETL) to obtain such narrow response?

At least for the studied active materials, it is not possible to obtain narrow response without ETL since no n-doping effect is involved. To prove that, we fabricated the ETL-free device (ITO/PCE10: COTIC-4F (500 nm)/MoO₃/Ag) and a broadband EQE was obtained. In the revised manuscript, we combined the data with other EQE curves in Fig. 2a. Please see the Fig. 2a in the revised version, and the figure caption has been revised accordingly.

Fig 2. The role of electron transporting layers (ETL) in manipulating EQE shapes. (a) ETL dependent EQE spectra in an inverted 500 nm-thick PCE10:COTIC-4F photodiode. For the ETL-free (grey dotted line) inverted device, the ITO substrate was not treated by UV Ozone since it would increase the energy barrier for electron extraction.

In the revised manuscript (page 7):

“ As a comparison, a ZnO sol-gel ~~ETL electron transporting layer based~~ and ETL-free devices were fabricated and similar broadband EQEs as for PFN-Br ~~was~~ were observed ”

Reviewer #2 (Remarks to the Author):

The manuscript by Liu et al. entitled “Protonated amine-interlayer induced n-type doping of polymer:nonfullerene blends for efficient narrowband near-infrared photo-detection” has presented a new detector design to achieve narrowband near-infrared response. The design involves incorporating amine interlayers near the anode. This approach is novel and less materials dependent and have been shown to work for 3 different blends, demonstrating the generalizability of this concept. While the device performance is excellent, there are some clarification requests as listed below:

1. The paper has shown that doping concentration is significant different with the use of PEIE and PDIN interface. Would there be a knob to change the doping concentration more precisely in the future,

Yes, there is. By quantitatively controlling the amount of PDIN in the NFA-blend, the n-doping concentration can be manipulated more precisely. We did the experiment of adding PDIN (8mg/mL, prepared in Chloroform+0.5 vol% CH₃COOH) into the PCE10: COTIC-4F (40 mg/mL) solution with a volume ratio of 1:20, and n-doping induced spectral narrowing was observed as shown by the EQE spectrum below (and Fig.S22a in the revised SI). However, PDIN tends to aggregate in the blend even when added in small quantities, ultimately altering the morphology of the solid-state blend film. As a result, the measured dark current under reverse bias is increased by more than four orders of magnitude (Fig.S22b). In future work, the electron-donating interfacial material should be carefully selected to ensure an appropriate n-doping density without significantly altering the blend morphology and charge transporting properties.

In the revised manuscript (page 12):

“The measured EQEs correspond to a larger w of ~ 250 nm, confirming the weaker doping of PEIE as compared to PDIN (see N_t in Table S1). A more precise control of the n-type doping concentration can be realized by adding a small, known amount of such electron-donating interfacial material in the NFA blend. Care must however be taken to ensure that the n-dopant does not significantly alter the blend morphology and charge transport properties, otherwise the dark current of the resulting narrowband OPD increases dramatically (Fig. S22).”

Figure S22. Measured EQE (a), and light and dark J - V curves (b) of the ITO/PDIN-doped PCE10:COTIC-4F (500 nm)/MoO₃/Ag device. The blend solution was prepared by adding PDIN (8mg/mL, prepared in Chloroform+0.5 vol% CH₃COOH) into the PCE10: COTIC-4F (40 mg/mL, CB+2%vol CN) solution with a volume ratio of 1:20. The increased dark current under reverse bias compared to Fig. 5a, is mainly ascribed to the PDIN aggregates induced morphology change of the solid-state blend film.

so the FWHM of the photoresponse can be further reduced? Or, from the simulations, can we extrapolate how N_t or depletion width in the supplemental table S1 will change the FWHM in EQE?

Besides the doping concentration or the depletion width, the FWHM of the photoresponse is more strongly dependent on the thickness of the photoactive layer. For a typical 500 nm-thick blend, a higher doping concentration (N_t) only slightly reduces the FWHM of the photoresponse. Please see the simulated results, shown in revised Fig.S5a and b, while increasing active layer thickness gradually reduces FWHM to ~30 nm if an 1100 nm-thick blend is applied (see revised Fig.S5 c-f).

To clarify, we add relevant discussion in the revised manuscript (page 6).

“EQEs in Fig. S7). We note that for a typical 500 nm-thick junction, a further decrease in the width of the SCR (w) or a higher n-doping concentration does not further reduce the FWHM. However, increasing the active layer thickness does (see FWHM of the simulated narrowband spectra as function of w and blend thickness in Fig.S5 c-f). A FWHM of ~30 nm for PCE10:COTIC-4F narrowband OPD is theoretically achievable for an 1100 nm-thick blend.”

Figure S5. (a) TMM computed spatially dependent absorption profiles in the device structure of ITO (135 nm)/PDIN (15 nm)/PCE10: COTIC-4F (500 nm)/MoO₃ (15 nm)/Ag (100 nm) with various widths of space charge region near the anode contact. (b) FWHM of the calculated narrowband response in the NIR as a function of the n-doping concentration (N_t) or the width of the SCR (w), where $w = \sqrt{\frac{2\epsilon_0\epsilon_r(V_{bi}-V)}{qN_t}}$. (c) and (e) are calculated thickness-dependent absorption spectra of PCE10: COTIC-4F and PCE10:PC₇₁BM with a fixed SCR width of 150 nm, respectively. (d) and (f) are the

corresponding plots of FWHM of the resulting narrowband responses in the NIR region as function of the blend thickness.

2. In the inset of Fig 4a, there is a dip in the noise spectra in the beginning. Would the authors explain why it is the case? It is not typical and opposite to the usual $1/f$ trend.

We thank the reviewer for this comment. To address the reviewer's concern, we have thoroughly revisited the experimental noise current density (NSD) spectra measurements and found the dips in noise spectra at low frequency to be related to the used current pre-amplifier when operating at high gain and low frequency. We would like to note that the specific detectivities were calculated based on noise currents obtained from NSD spectra at high frequency (i.e., >500 Hz) and were therefore not affected by the dips in noise spectra. Based on these findings, we carefully repeated the NSD measurements using an electrical noise-minimized and current pre-amplifier-free setup. As a result, the noise spectra are now obtained from fast Fourier transform (FFT)-based transient dark current measurements. The calculated NSD spectra show no dips at low frequency. noise currents obtained via FFT-based transient dark current are consistent with values obtained via direct NSD spectra measurement at higher frequencies. The experimental method section, Fig. 4a (Fig.5a in revised manuscript), and Fig. S19 (Fig.S24 in the revised Supporting Information) have been revised accordingly.

Figure S24. Noise spectral density (NSD) spectra plotted as a function of frequency of PCE10:FOIC (red), PCE10:SiOTIC-4F (black), and PCE10:COTIC-4F (blue) obtained at (a) 0 V, (b) -0.5 V, and (c) -1 V applied bias voltage. Noise currents obtained via FFT-based transient dark current here are consistent with values obtained via direct NSD spectra measurement at higher frequencies. (d) Noise currents as obtained *via* NSD (filled symbols) and calculated noise (dashed lines) as $i_{\text{noise}} = \sqrt{2qi_d\Delta f + \frac{4k_B T\Delta f}{R_{\text{shunt}}}}$ (where q is the elementary charge, k_B the Boltzmann constant, T the temperature, i_d the dark current, R_{shunt} denotes the shunt resistance, and Δf is the electrical bandwidth) plotted as function of bias voltage, and compared for the three narrowband photodetectors from a-c. While the horizontal solid line marks the noise floor of the Semiconductor Device Analyzer (Keysight B1500A), open symbols

correspond to the calculated thermal noise of the three photodetectors at zero applied bias voltage as

$i_{\text{noise}} = \sqrt{\frac{4k_B T \Delta f}{R_{\text{shunt}}}}$. Here, the electrical bandwidth was set to $\Delta f = 1$ Hz. The corresponding NSD- and J - V - i_d values coincide at high reverse bias voltage due to the suppression of parasitic shunt current effects dominant at small reverse voltages close to 0 V.

In the revised manuscript (Methods, Page 17):

“**Noise and frequency response measurement.** Noise spectral density (NSD) spectra were obtained by recording the dark currents as a function of time and applying a fast Fourier transform (FFT) algorithm. A semiconductor device analyser (Keysight B1500A) was used for dark current measurements at different bias voltages. The photodetectors were mounted in an electrical-shielded sample holder (Faraday cage) to minimize electrical noise. The frequency response speeds of the narrowband NIR photodetectors were recorded by using a Network Analyzer (Keysight E5071C ENA Vector Network Analyser) in transmission mode modulating an Oxixus L6Cs laser ($\lambda_{\text{exc}} = 520$ nm) with frequencies ranging from 1 Hz to 1 MHz.”

3. For the color map in Fig 1a, is the color map on the left in arbitrary units? The scale there is somewhat different from the Supplement calculations in Fig S5 where the calculation absorption in a.u. are ~ 0.2 , but the color maps in Fig 1a and supplemental Fig. S4 is 10x smaller in 0.02.

Yes, Fig. 1a and Fig. S4 are the contour plots of the spectrally and spatially resolved energy absorption in arbitrary units. While, Fig. S5 is the absorption calculated by integrating all the absorption from position 0 to 500 nm (the whole active layer) at each wavelength. That's why the a.u number in Fig.S5 is much larger than Fig.1a and Fig. S4.

4. There are a lot of acronyms, making it somewhat difficult to follow, and there is one acronym SCCN that was undefined on p10, but I think it's spectral charge collection narrowing? Anyways, it might be more readable if some acronyms are spelled out in words if possible.

We are sorry for forgetting to remove the acronym SCCN in the submitted version, which was defined as space charge collection narrowing in the earlier draft. We thought it would not be suitable and decided to remove it. Please see the revised manuscript (page12).

“Having established the mechanism of n-doping enabled ~~SCCN~~ spectral narrowing in the NIR photodiodes, we now evaluate their photo-detection performance.”

We also checked all other acronyms in the text and make sure all of them are well defined in the revised text. For example, “NIR organic photodetectors (NIR-OPD), and electron transport layer (ETL).”

Reviewer #3 (Remarks to the Author):

The manuscript of Liu and coworkers reports narrowband NIR organic photodetectors with a full-width-at-half-maximum of around 50 nm and specific detectivities of $>10^{12}$ Jones. These results are achieved by the n-doping of the active layer induced by protonated amine-based electron transporting layers. The authors pushed further the charge collecting narrowing method developed by Armin et al., and found that a small doping concentration creates a space charge region with widths of 150-250 nm, which leads to narrowband NIR-detection at the absorption onset of the active layer. I have some concerns about the n-doping effect, which are listed below. However, I believe these findings are important for the community and can help the development of NIR OPD. For this reason, I support the publication of this manuscript in nature communications upon replies to the following questions.

1) The authors claimed that the n-doping effect and consequentially the SCR is induced by the ETL. The authors supported this statement by indicating that trap-assisted recombination is not playing a role by carrying out light-intensity JVs (Figure S8). I believe that further measurements are necessary to exclude trap-assisted recombination, i.e. charge extraction and transient photovoltage measurements.

Measurements of V_{oc} vs intensity are a reliable way to study trap-assisted recombination in organic photovoltaics. Indeed, this is what we have done and shown in Figure S8 (Figure S12 in the revised supplementary information). TPV and charge extraction measurements do not contain more information on trap assisted recombination, and moreover, interpretation of those measurements is difficult for doped layers. Up to now, when those measurements are done in the literature, no doping is assumed. This makes it really hard to draw more meaningful conclusions on surface trap assisted-recombination from such measurements. We therefore refrained from adding such measurements to the manuscript. We have however, in this revised manuscript, considerably strengthened the evidence for n-doping being the ETL (please see answers to previous referee questions).

2) In addition, a common method to study doping is EPR. I strongly recommend the authors to carry out this measurement.

We agree with the review that EPR is a good tool to study the doping for organic semiconducting materials, but we and our collaborators do not have access to such a setup at this moment. However, to see the doping effect and calculate the doping density of the device, capacitance-voltage measurement is more reliable and meaningful (refer to Fig. 2c and Fig. S20 containing this measurement). In addition, We cite relevant papers that show EPR data for the self-doping n-type ETL and n-doping NFA blend films, to further support the interlayer can induce an n-type doping of NFA. (see ref.50-55 in the revised manuscript).

3) The morphology of the active layer can be drastically influenced by the layer below. I suggest performing contact angle measurements for active layers coated on different ETL to exclude this variable. Following the reviewer's suggestion, we performed contact angle measurement on PCE10: COTIC-4F blend films coated on the studied PFN-Br, ZnO, PEIE and PDIN interfacial layers. As shown in the revised Fig. 8a, all samples have a very similar contact angle (CA, water) closed to that of neat polymer film of 99°, indicating a donor polymer-rich top surface. In addition, we also performed atomic force microscope (AFM) measurements on such different ETL coated blend films, shown in Fig S8b. Again, no obvious difference in root mean square (RMS) roughness and significant morphology change is observed amongst those samples. Then, it would be safe to exclude severe morphological changes of the active layer when different ETLs are applied.

Figure S8. (a) Static contact angle measurements with water on top of the neat PCE10 polymer, COTIC-4F NFA and ETL-coated PCE10: COTIC-4F blend films (500 nm). All ETL-coated blend films have a very similar contact angle (CA, water) closed to that of neat PCE10 film of 99°, indicating a polymer-rich top surface. (b) Topography atomic force microscope (AFM) images. No obvious difference in root mean square (RMS) roughness and significant morphology change is observed amongst the studied samples.

In the revised manuscript (Page 7):

“Note that the morphology of the active layer is negligible influenced by the electron-transporting interfacial layer below, which is confirmed by atomic force microscope (AFM) and static water contact angle measurements on the different ETL-coated blend films (Fig. S8).”

4) In some figures, the active layer thickness is missing.

We have checked all the figures in the main text, and the active layer thickness in Fig.5 has been added. Please see the revised figure caption for Fig. 5a. (page 24 in the revised manuscript)

“(a) Dark current density-voltage (J - V) characteristics of PCE10: FOIC (600 nm), PCE10: SiTOC-4F (600 nm) and PCE10: COTIC-4F (500 nm) narrowband NIR-OPDs.”

REVIEWERS' COMMENTS

Reviewer #1 (Remarks to the Author):

The performed additional experiments addressed the previous concerns. The results enabled the conclusion more convincing.

Reviewer #2 (Remarks to the Author):

The revisions have satisfactorily addressed the reviewers' comments.

Reviewer #3 (Remarks to the Author):

The authors replied to all my comments/concerns, motivating their conclusions and adding novel data as requested. I therefore, recommend to accept the manuscript.

Reviewer #1 (Remarks to the Author):

The performed additional experiments addressed the previous concerns. The results enabled the conclusion more convincing.

We thank the reviewer for his/her positive feedback on our manuscript.

Reviewer #2 (Remarks to the Author):

The revisions have satisfactorily addressed the reviewers' comments.

We thank the reviewer for his/her positive feedback on our manuscript.

Reviewer #3 (Remarks to the Author):

The authors replied to all my comments/concerns, motivating their conclusions and adding novel data as requested. I therefore, recommend to accept the manuscript.

We thank the reviewer for his/her positive feedback on our manuscript.